# A lab-based test of the gravitational redshift with a miniature clock network

Xin Zheng[1,3], Jonathan Dolde[1,3], Matthew C. Cambria [1], Hong Ming Lim[1] & Shimon Kolkowitz [1,2] ✉

Einstein's theory of general relativity predicts that a clock at a higher gravitational potential will tick faster than an otherwise identical clock at a lower potential, an effect known as the gravitational redshift. Here we perform a laboratory-based, blinded test of the gravitational redshift using differential clock comparisons within an evenly spaced array of 5 atomic ensembles spanning a height difference of 1 cm. We measure a fractional frequency gradient of $[-12.4 \pm 0.7_{(stat)} \pm 2.5_{(sys)}] \times 10^{-19}$/cm, consistent with the expected redshift gradient of $-10.9 \times 10^{-19}$/cm. Our results can also be viewed as relativistic gravitational potential difference measurements with sensitivity to mm scale changes in height on the surface of the Earth. These results highlight the potential of local-oscillator-independent differential clock comparisons for emerging applications of optical atomic clocks including geodesy, searches for new physics, gravitational wave detection, and explorations of the interplay between quantum mechanics and gravity.

Einstein's theory of general relativity[1] has thus far been consistent with every experimental test performed[2], including classical[3,4], modern[5,6] and strong field cosmological tests[7–9]. However, despite the successful integration of special relativity and quantum mechanics as quantum field theory, there is currently no theory that successfully unifies general relativity with quantum mechanics. This motivates continued experimental tests at new length scales, and suggests that performing precision tests of general relativity with quantum systems may offer a way to explore the interplay between general relativity and quantum mechanics[10–12].

The gravitational redshift is a central prediction of general relativity. Thanks to rapid advancements in their stability and accuracy[13–19], atomic clocks have now enabled tests of the gravitational redshift over a wide range of length scales[20–25]. Recent tests of the gravitational redshift include a frequency comparison between two single-ion clocks with one of the clocks elevated by 30 cm[21], comparisons between terrestrial clocks and microwave atomic clocks in eccentric orbits which have produced the strongest limits on deviations from the expected redshift[22,23], a frequency comparison between two synchronously linked portable [87]Sr optical lattice clocks with a 450 m height difference at the Tokyo Skytree tower[24] resulting in the most precise

terrestrial constraint on deviations from the gravitational redshift at the $10^{-5}$ level, and a recent in-situ synchronous frequency gradient measurement across a millimeter-scale atomic ensemble with an unprecedented differential precision of $7.6 \times 10^{-21}$[25].

For two otherwise identical clocks experiencing the same gravitational field with a height difference $\Delta h$, their frequency difference $\delta f$ due to the gravitational redshift is given by

$$\frac{\delta f}{f} \approx \frac{g \Delta h}{c^2} \qquad (1)$$

where $f$ is the clock frequency, $c$ is the speed of light, and $g$ is the gravitational acceleration. Near the surface of Earth, this amounts to a fractional frequency shift of $1.1 \times 10^{-18}$ per centimeter of vertical displacement. With optical clocks now reaching instabilities and inaccuracies at the level of $10^{-18}$ and below[15–17,26,27], they are becoming a sensitive probe of the point to point geopotential at the sub-centimeter scale, where they are expected to complement other methods of geodesy[15,28–35]. For example, a blinded comparison between two independent Yb optical lattice clocks was recently performed with accuracy, instability and reproducibility all at the level

[1]Department of Physics, University of Wisconsin-Madison, Madison, WI 53706, USA. [2]Department of Physics, University of California, Berkeley, CA 94720, USA. [3]These authors contributed equally: Xin Zheng, Jonathan Dolde. ✉e-mail: kolkowitz@berkeley.edu

required to resolve sub-cm height differences[15]. In addition, the frequency gradient due to the gravitational redshift across a single millimeter-scale atom ensemble was recently observed using Rabi spectroscopy of $^{87}$Sr without the use of a blinding offset and taking advantage of an 8-mHz linewidth clock laser[25].

Emerging clock applications such as relativistic geodesy require transportable optical clocks, which currently have poorer stabilities and accuracies than those of laboratory-based clocks[32,34,36]. State-of-the-art laboratory-based optical clocks often make use of bulky and immobile reference cavities with second-scale coherence times[37–39] in order to achieve lower levels of clock instability and to aid in rapid systematic evaluation, limiting deployment in the field. It has recently been demonstrated that differential measurements between single ions[40] and neutral atom ensembles[41–43], as well as differential spectroscopy by phase-coherently linking a zero-dead-time optical lattice clock and a single ion clock[44], allow interrogation times beyond the limit set by the local oscillators, opening up the prospects for future applications with transportable or space-based clocks[45–49].

In this work, we perform a local-oscillator independent, blinded test of the gravitational redshift at the sub-centimeter scale using a spatially multiplexed optical lattice clock[43] consisting of an array of $^{87}$Sr atom ensembles trapped in a vertical, one-dimensional (1D) optical lattice (Fig. 1a). We prepare five atomic ensembles equally spaced by 2.5 mm, spanning a total height difference of 1 cm. Synchronous differential comparisons are performed between the five ensembles, resulting in ten unique pairwise clock comparisons recorded simultaneously, including four pairs at 2.5 mm, three pairs at 5.0 mm, two pairs at 7.5 mm, and one pair at 1.00 cm height difference. The gravitational redshift is tested by mapping out the frequency differences between each ensemble pair as a function of height difference. We measure a fractional frequency gradient of $[-12.4 \pm 0.7_{(stat)} \pm 2.5_{(sys)}] \times 10^{-19}$/cm, consistent with the expected

gravitational redshift gradient of $-10.9 \times 10^{-19}$/cm. Our result constrains deviations from the redshift predicted by general relativity to $0.13 \pm 0.23$ for mm to cm scale height differences.

## Results

### Testing the gravitational redshift with a miniature clock network

The principles and basic operation of the multiplexed optical lattice clock used in this work were first described and experimentally demonstrated in ref. 43. In that work we demonstrated differential clock comparisons between atom ensembles with fractional frequency imprecision below $1 \times 10^{-19}$. However, there are a wide range of potential sources of differential frequency shifts at the mHz level, so the exact origin of the frequency differences we measured was not known, and the contributions from the relativistic gravitational redshift could not be independently extracted. In this work, we leverage several key improvements to our apparatus and experimental procedure that have been made since our prior work, and perform a full systematic evaluation of all potential sources of differential frequency shifts at the $10^{-19}$ level, enabling a blinded test of the gravitational redshift with mm-scale sensitivity to gravitational potential differences.

Our major improvements include employing a deeper initial optical lattice ($130E_{rec}$ trap depth, where $E_{rec}$ is the recoil energy of a lattice photon) during atomic ensemble loading and in-lattice cooling provide larger atom numbers (>2000 atoms per ensemble) with reduced atomic temperatures both axially (>99% occupancy in the lattice ground band) and radially (<200 nK). Synchronized Ramsey measurements are subsequently performed at a shallower operational lattice depth ($u_{op}$) of $15E_{rec}$. Combined with common-mode rejection of clock laser noise, we achieve 32 s atom-atom coherence times, more than 300 times longer than our measured atom-laser coherence time

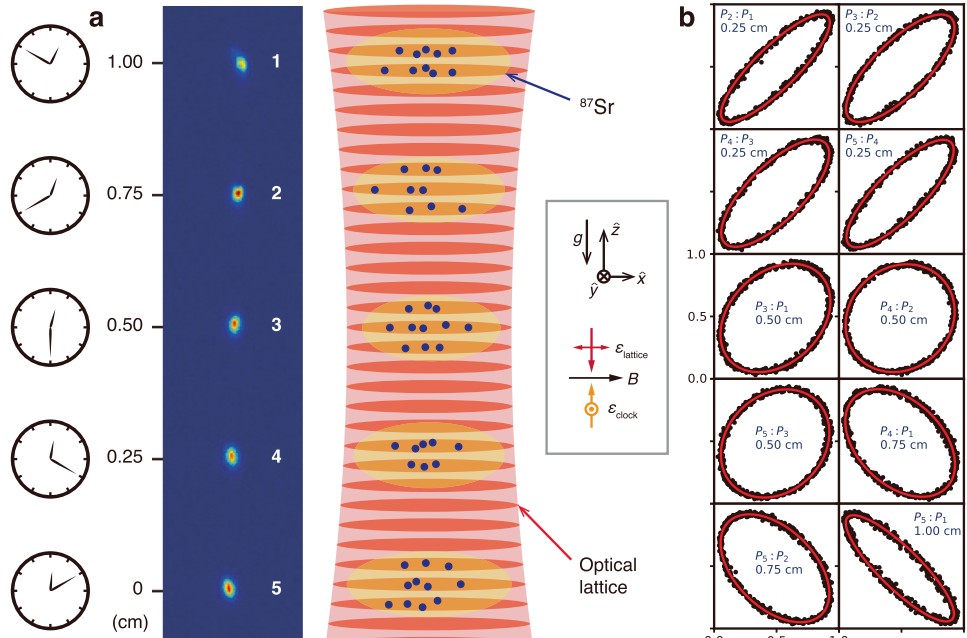

**Fig. 1 | Experimental system and synchronous differential clock comparisons. a** A representative camera image of a spatially multiplexed array with five ensembles of $^{87}$Sr atoms (indexed 1–5 from top to bottom) trapped in a vertical 1D optical lattice for differential clock comparisons. The spacing between neighboring ensembles is 0.25 cm, spanning a total height difference of 1 cm. Due to the gravitational redshift, clocks at a higher gravitational potential are predicted to tick faster than clocks at a lower potential. The gray box shows the orientations of the applied bias magnetic field (**B**), and the lattice and clock laser polarizations ($\epsilon$). **b** A

representative outcome from synchronous Ramsey spectroscopy on the $|^1S_0, m_F = +5/2\rangle \leftrightarrow |^3P_0, m_F = +3/2\rangle$ clock transition with a 10 s free precession time using five atomic ensembles, resulting in ten pairwise clock comparisons. In each plot, the excitation fractions of ensemble $j$ are plotted against the excitation fractions of ensemble $i$ ($P_j$:$P_i$, where $j > i$), tracing out an ellipse which is fitted to in order to extract the frequency difference between that pair. The frequency difference is dominated by the first-order differential Zeeman shift, which is rejected when averaging transitions between opposite spin states (Supplementary Note 3A).

of roughly 100 ms (Supplementary Note 2). This enables differential instabilities below $1 \times 10^{-17}/\sqrt{\tau}$ for all the ensemble pairs, a factor of 3 reduction in instability over our previous work ($3 \times 10^{-17}/\sqrt{\tau}$ with six ensembles). In addition, we now suppress the residual magnetic field gradient along the lattice axis by a factor of more than ten, reducing systematic uncertainties arising from Zeeman shifts. These advances allow us to rapidly evaluate the differential systematic shifts at the $10^{-19}$ level due to environmental perturbations and thus perform a precision test of the gravitational redshift. To the best of our knowledge, our work represents the first complete systematic evaluation of differential frequency shifts at the $10^{-19}$ level making use of synchronous Ramsey spectroscopy. This technique, which unlike differential Rabi spectroscopy can be used to probe well beyond the coherence time of the local oscillator, enables a new modality of precision measurement with optical lattice clocks where both the achievable accuracy and precision are now unbounded by the quality of the local oscillator.

For our measurements, we utilize synchronous Ramsey spectroscopy in conjunction with spatially resolved fluorescence imaging to probe the clock transition along the ensemble array. The optical lattice is operated near the magic wavelength where the differential polarizability between the ground ($^1S_0, g$) and clock ($^3P_0, e$) states is zero[50,51]. We probe with an interleaved sequence using the magnetically insensitive $|g, m_F = \pm 5/2\rangle \leftrightarrow |e, m_F = \pm 3/2\rangle$ clock transitions[18], where $m_F$ is the projection of total angular momentum along the quantization axis defined by the bias magnetic field. Taking the average of the clock transitions with opposite sign nuclear spin states rejects first-order Zeeman shifts and vector AC Stark shifts. The differential phase ($\phi_d$) for each ensemble pair is extracted through least squares ellipse fitting, and is related to the differential frequency ($\delta f$) for the pair through $\phi_d = 2\pi \delta f T_R$, where $T_R$ is the Ramsey free evolution time. A representative outcome from clock interrogation on the $|g, m_F = +5/2\rangle \leftrightarrow |e, m_F = +3/2\rangle$ transition in shown in Fig. 1b, where ($\delta f$) is dominated by the first-order Zeeman shift (Supplementary Note 3A).

The differential frequency between each atomic ensemble pair includes a contribution from the gravitational redshift as well as other frequency shifts arising from differences between the two ensembles and their environments, necessitating an evaluation of potential sources of systematic effects. To avoid possible bias towards the expected result, we adopt a blinded measurement protocol. A blinded offset gradient was randomly drawn from a range of $\pm 5 \times 10^{-18}/$cm, roughly 10 times the size of the expected redshift gradient, and is automatically added to the extracted differential phase by our data analysis code following the ellipse fitting step. This blinded offset was not known to the authors during systematic evaluation and data taking. The blinded value of the measured frequency gradient across the array, the corrections for systematic shifts, and the systematic and statistical uncertainties were all finalized prior to the removal of the blind.

**Systematic effects and error budget**
The results of the full systematic evaluation are listed in Table 1. The procedure for measuring the contribution of each potential systematic is discussed in Supplementary Note 3. Several effects dominate the extracted differential frequency and the corresponding systematic uncertainty, and we highlight them here. First, atomic interactions due to $p$-wave collisions between on-site atoms lead to a frequency shift that scales linearly with atomic density[52–54]. In our differential measurement, the density shift is suppressed by a factor of roughly 10 compared to the absolute shift by balancing the number of atoms loaded into each ensemble. By intentionally varying the atom number differences, the differential density shift for a symmetric pair (2, 4) at $u_{op}$ is evaluated to be $-0.7(1) \times 10^{-19}$ per 100 atom number difference (Fig. 2a). Due to the Gaussian nature of the lattice beam, we observe a minor trap volume dependence of the density shift, which is

**Table 1 | Fractional frequency gradients and corresponding uncertainties**

| Sources | Gradient ($\times 10^{-19}$/cm) | Uncertainty ($\times 10^{-19}$/cm) |
|---|---|---|
| BBR[a] | −15.7 | 1.5 |
| Lattice light | −11.8 | 1.2 |
| Density[b] | – | 1.0 |
| 2nd order Zeeman[a] | −95.3 | 1.0 |
| Probe Stark | 0 | 0.5 |
| DC Stark | 0 | 0.1 |
| Ellipse fitting[b] | – | 0.5 |
| Total systematic correction | +122.8 | 2.5 |
| Statistical gradient | −135.2 | 0.7 |
| Corrected gradient | −12.4 | 2.6 |
| Expected redshift gradient | −10.9 | <0.1 |

Uncertainties are quoted as 1σ standard deviations. For each systematic effect, more discussion can be found in Supplementary Note 3.
[a]The BBR shift and second-order Zeeman shift are corrected for each measurement, here the weighted averaged values across all 14 measurements are listed
[b]The density shift and bias error from ellipse fitting are corrected for each pairwise clock comparison individually, and do not depend linearly on spatial separation.

accounted for in our evaluation by independently extracting the density shift as a function of atom number difference for each pairwise comparison (Supplementary Note 3B). In each measurement run, the corresponding differential density shifts are corrected for each ensemble pair individually.

The largest systematic shift in our system is the second-order Zeeman shift due to the background magnetic field gradient (~1.5 mG/cm). The splitting of the transitions with opposite sign nuclear spin states provides a measurement of the magnetic field difference between each ensemble pair, while the overall magnitude of the applied bias magnetic field (~5.5 G) is measured independently using a more magnetically sensitive transition. This bounds the uncertainty from the second-order Zeeman gradient to below $1 \times 10^{-19}$/cm, limited by uncertainty in the second order Zeeman coefficients for the clock transition (Fig. 2b and Supplementary Note 3A).

The frequency shift due to black body radiation (BBR) is the dominant source of systematic uncertainty for many room-temperature optical clocks[14–17]. In our system, uncertainty arises due to temperature gradients in the surrounding environment. Because the ensembles are arrayed vertically, the primary contribution comes from differences in temperature between the top and bottom recessed viewports of the science chamber, which are the closest surfaces to the atoms. To evaluate the effect of this BBR gradient, we intentionally introduce a thermal gradient by varying the temperature difference between the two viewports by up to ±1 K. Mapping out the resulting differential frequency shifts of the ten ensemble pairs, we measure a linear BBR gradient sensitivity of $-4.2(1) \times 10^{-18}$/cm per 1 K difference (Fig. 2c and Supplementary Note 3C). This results in a weighted average BBR gradient of $-15.7(1.5) \times 10^{-19}$/cm under normal operating conditions, which is determined from the temperature differences recorded with calibrated temperature sensors that were continuously monitored during data taking. This measurement also highlights the application of differential comparisons between spatially multiplexed ensembles for BBR gradient evaluation at the $10^{-19}$/cm level, which will be important for pushing the accuracy of room temperature optical lattice clocks below the $10^{-18}$ level.

AC Stark shifts from lattice light contribute to the fractional frequency uncertainty of state-of-the-art optical lattice clocks at the low $10^{-18}$ level[50,51]. For differential comparisons between ensembles within a single shared optical lattice, this is significantly suppressed. A differential lattice AC Stark shift between two ensembles is caused by the

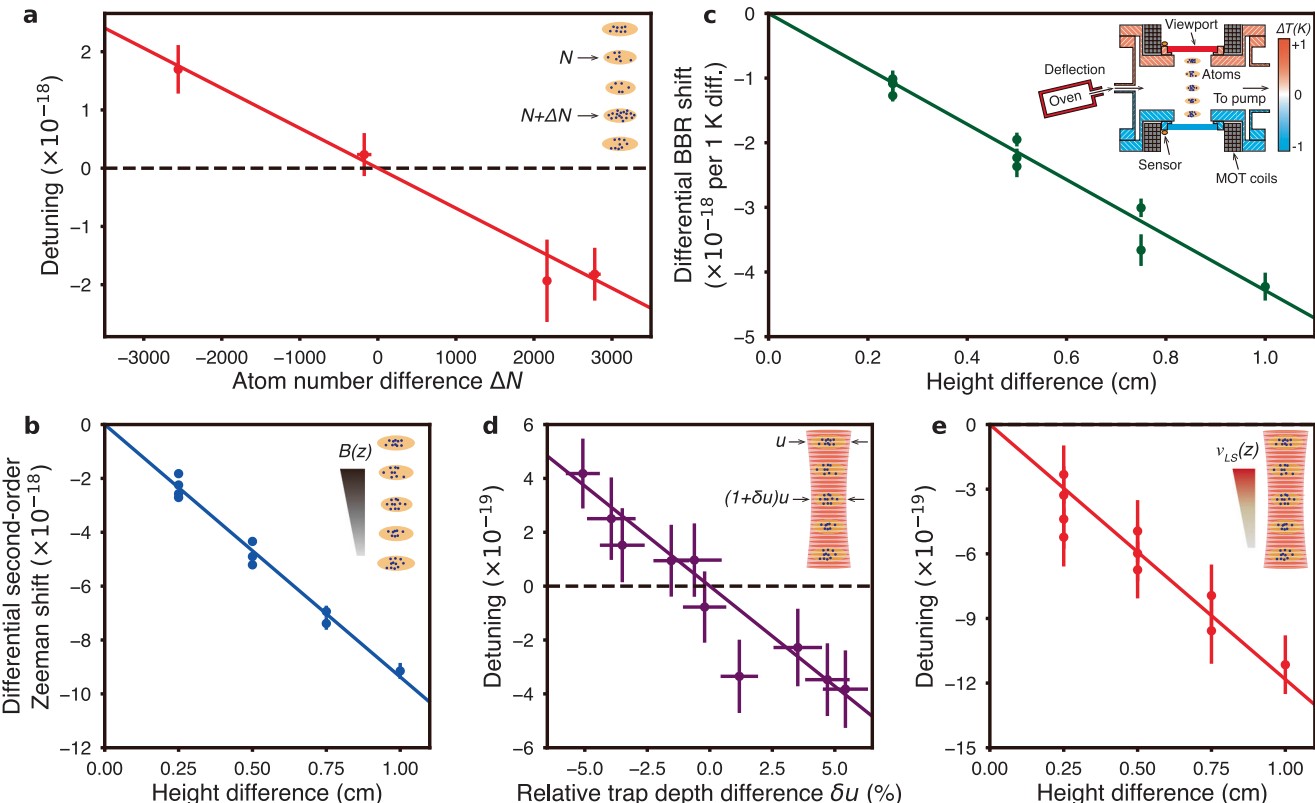

**Fig. 2 | Sources and characterization of primary systematic shifts.** The error bars correspond to 1σ standard deviation. **a** The differential density shift at $u_{op} = 15E_{rec}$ for a symmetric pair (2, 4) is evaluated by varying the atom number difference (ΔN). A linear fit yields a shift of −0.7(1) × 10⁻¹⁹ per 100 atom number difference. **b** Evaluation of the second-order Zeeman gradient arising from the magnetic field gradient ($\partial B/\partial z \approx 1.5$ mG/cm). A linear fit yields a gradient of −95.3(1.0) × 10⁻¹⁹/cm. **c** Characterization of the BBR shift due to thermal gradients across the vacuum chamber. The inset is an illustration of the science chamber. To evaluate the BBR effect, a thermal gradient is introduced by varying the temperature difference between top and bottom viewports by up to ±1 K. A linear fit yields a BBR sensitivity of −4.2(1) × 10⁻¹⁸/cm per 1 K difference in our system. **d** Correlations between relative trap depth difference (δu) and differential lattice light shifts after subtraction of the residual spatial light shift gradient. u and (1 + δu)u correspond to the absolute trap depths for the ensemble pairs. **e** Evaluation of lattice light shift gradient at $u_{op} = 15E_{rec}$ after removing contributions from δu shifts. A linear fit yields a gradient of −8.0(1.1) × 10⁻²⁰/$E_{rec}$/cm.

relative trap depth difference δu arising from the lattice beam profile, and scales linearly with δu to first-order (Supplementary Note 3D). In conjunction with the multiplexed ensemble technique, we can rapidly map out both δu and the differential light shifts. In doing so, we also observe a residual spatial light shift gradient of −8.0(1.1) × 10⁻²⁰/$E_{rec}$/cm, which depends linearly on the lattice depth and the spatial separation between ensembles, and does not depend on δu. We believe this gradient is likely due to the differential tensor Stark shift arising from slight variations in the orientation of the magnetic field vector across the ensemble array. This is supported by the observation of a differential vector Stark gradient of −2.5(2) × 10⁻¹⁸/$E_{rec}$/cm in our system (Supplementary Note 3D). Regardless of the exact origin of the spatial light shift gradient, we are able to measure and account for it by varying the lattice depth. Subtracting off this residual spatial gradient, we observe correlations between δu and the remaining shifts (Fig. 2d), as expected. This allows us to extract the operational lattice detuning from the effective magic wavelength[25,51], and to independently evaluate the lattice light shift upon removal of the δu shifts (Fig. 2e).

### Data analysis and interpretation

We performed 14 blinded measurements of the gravitational redshift under normal operating conditions over a 3-week data taking campaign. In each measurement run (ranging in duration from 1 to 4 h), the frequency gradient is determined by fitting a linear slope to the ten measured differential frequencies as a function of the pairwise height differences, after taking into account systematic corrections such as

ellipse fitting bias and density shift. In order to account for correlations that arise between clock comparison pairs that share the same clock, the covariance between the pairwise comparisons is included in the error estimation (see Methods for details). Corrections for spatially varying systematics such as the lattice light shift, BBR shift, and second-order Zeeman shift are applied to the measured gradient. Upon removal of the blinded offset gradient, we find a weighted mean fractional frequency gradient of $[-12.4 \pm 0.7_{(stat)} \pm 2.5_{(sys)}] \times 10^{-19}$/cm (where 'stat' and 'sys' indicate statistical and systematic uncertainties, respectively), consistent with the expected redshift gradient of −10.9 × 10⁻¹⁹/cm within 1σ total uncertainty (Fig. 3a). The statistical uncertainty is scaled up by the square root of the reduced $\chi^2$ statistic, $\chi^2_{red} = 1.16$. Our measurement is inconsistent with the hypothesis that there is no gravitational redshift on the surface of the Earth for mm-cm scale height differences at a confidence level of 4.9σ.

Deviations from the gravitational redshift predicted by general relativity can be parameterized by defining a modification parameter α

$$\frac{\delta f}{f} = (1 + \alpha)\frac{g\Delta h}{c^2} \qquad (2)$$

to first-order of the gravitational potential difference[2,24]. We constrain deviations from the predicted scaling by $\alpha = 0.13 \pm 0.23$ for millimeter to centimeter scale height differences. We note that while the most stringent constraints on α are at the 10⁻⁵ level[22,23], those measurements were performed at very different length scales, with height differences roughly a factor of 10⁹ times larger than those used here.

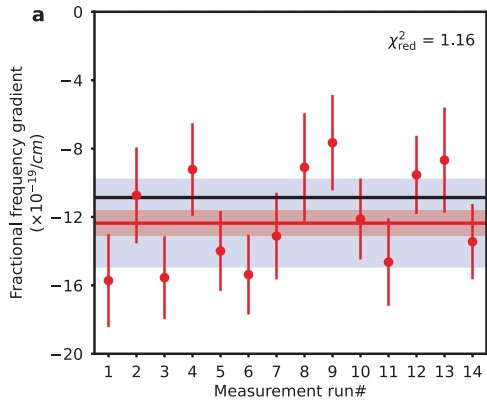
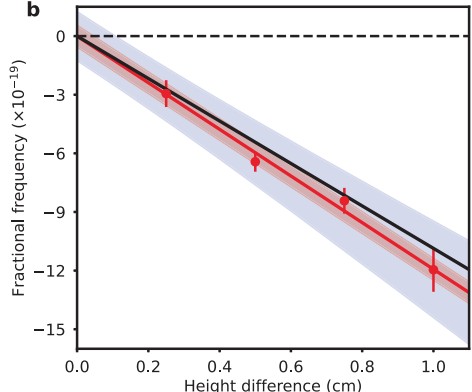

**Fig. 3 | Gravitational redshift measurements. a** The measured fractional frequency gradients after accounting for all systematic corrections over 14 data taking runs (red scatter points), each of duration ranges from 1 to 4 h. The weighted average (red solid line) yields a measured gradient of $[-12.4 \pm 0.7_{(stat)} \pm 2.5_{(sys)}] \times 10^{-19}$/cm, consistent with the expected redshift gradient (black solid line). Red (blue) area represents $\pm 1\sigma$ statistical (total) uncertainty, in which the statistical uncertainty is inflated by the square root of the reduced $\chi^2$ statistic, $\chi^2_{red} = 1.16$. The error bars correspond to $1\sigma$ standard deviation. **b** Mean differential frequencies as a function of height difference across all measurements (red scatter points), analyzed using the same data set as in (**a**). A linear fit (red solid line) yields a fractional frequency gradient of $(-11.9 \pm 2.6) \times 10^{-19}$/cm, again fully consistent with the expected redshift gradient (black solid line).

In the future it appears feasible to extend our measurement approach to a larger apparatus in order to achieve spatial separations between ensembles on the scale of roughly 1 m, a scale of laboratory atomic physics experiment that has already been demonstrated with atom interferometers[55], which would offer an increase in the magnitude of the redshift by a factor of 100× over the separations used in this work. In addition, we see a clear path to reducing the systematic uncertainty of our differential comparisons by more than one order of magnitude, which can be accomplished by mitigating density shifts via operating at "magic" excitation fractions (as was recently demonstrated within a single ensemble[25]) or through modifications to the lattice geometry[19], reducing the differential tensor lattice light shift by eliminating background magnetic field gradients, and mitigating the black-body radiation gradients through improved control of the thermal environment. When combined, such an experiment would promise constraints on $\alpha$ at the $10^{-4}$ level, indicating that laboratory-based tests of the gravitational redshift could soon be competitive with space-based tests[22,23] or tests with portable clocks[24].

As an alternative to the above analysis, we reanalyze the same experimental data by taking the weighted average of the differential frequencies of each ensemble pair from the 14 measurement runs, with systematic corrections applied individually to each pair. The weighted mean differential frequencies of ensemble pairs with the same height differences (0.25, 0.50, 0.75 and 1.00 cm) are shown in Fig. 3b. Through a final linear fit, we find a frequency gradient of $(-11.9 \pm 2.6) \times 10^{-19}$/cm, again fully consistent with the expected redshift. Extrapolating the spatially varying systematic uncertainties to 0 cm of separation, we find our gravitational redshift resolution to be 1.3 mm, dominated by systematic uncertainty due to the differential density shift, which could potentially be further reduced by incorporating the recent technique for density shift cancellation[54] in future work (Supplementary Note 3B).

## Discussion

The measurement of gravitational potential differences between clocks with sub-centimeter resolution is a major goal of relativistic geodesy[15,31–34]. In the preceding discussion and analysis we treated the height differences between each ensemble pair and the local gravitational acceleration $g$ as known, as we measure them independently, but did not a priori trust the gravitational redshift predicted by general relativity. From another perspective, our measurements can be viewed as a proof-of-principle demonstration of relativistic gravitational potential measurement with millimeter scale resolution. Taking as a given that the redshift is given by Eq. (1) and treating the ensemble array as a network of spatially distributed clocks with unknown height differences, we can extract the height ordering and relative height differences from the measured gravitational redshifts and $g$, which we measure independently. We find that we correctly assign the order of gravitational potential differences within the network, and that all of the extracted height differences are within 2 mm of the known values (Fig. 4). However, we note that we greatly benefit from the rejection of common-mode systematic shifts thanks to the ensembles sharing the same optical lattice and the same science chamber, which will not be possible when comparing two individual clocks at different geospatial locations. In addition, over a long baseline (>1 km), phase noise from the frequency transfer will not be common-mode and will limit the coherence times of the differential comparison. Therefore, while these results demonstrate that relativistic gravitational potential measurements with mm-scale height resolution are achievable in the lab over short spatial separations, considerable challenges must be overcome before they can be applied to relativistic geodesy at length scales of interest.

In a recent work[25], Bothwell and collaborators resolved the gravitational redshift across a single 1 mm atom ensemble. While there are aspects in common between this work and ref. 25, there are also several critical differences that set this work apart. First, we employed a blinded offset during data taking and systematic evaluation. Second, while Bothwell et al. made use of second-scale Rabi spectroscopy with an 8 mHz linewidth clock laser, we demonstrate comparable levels of differential stability and perform a full systematic evaluation at the $10^{-19}$ level by employing synchronous Ramsey spectroscopy with a Hz linewidth clock laser. This demonstrates that measurements of this kind need not be limited be the stability of the local oscillator. Third, we measure between spatially resolved ensembles using techniques that are likely more relevant to applications that require spatially separated clocks such as relativistic geodesy and gravitational wave detection. Finally, we also observe and characterize several systematic effects that were not observed by Bothwell et al., such as a black-body radiation gradient shift and differential tensor lattice light shift, likely in part because of the larger range of spatial separations used in our work.

In conclusion, we perform a blinded, precision test of the gravitational redshift on the sub-centimeter scale with five spatially multiplexed ensembles of $^{87}$Sr. We observe a gravitational redshift for millimeter to centimeter scale differences in height, and find that it is consistent with the expected general relativity gravitational redshift to

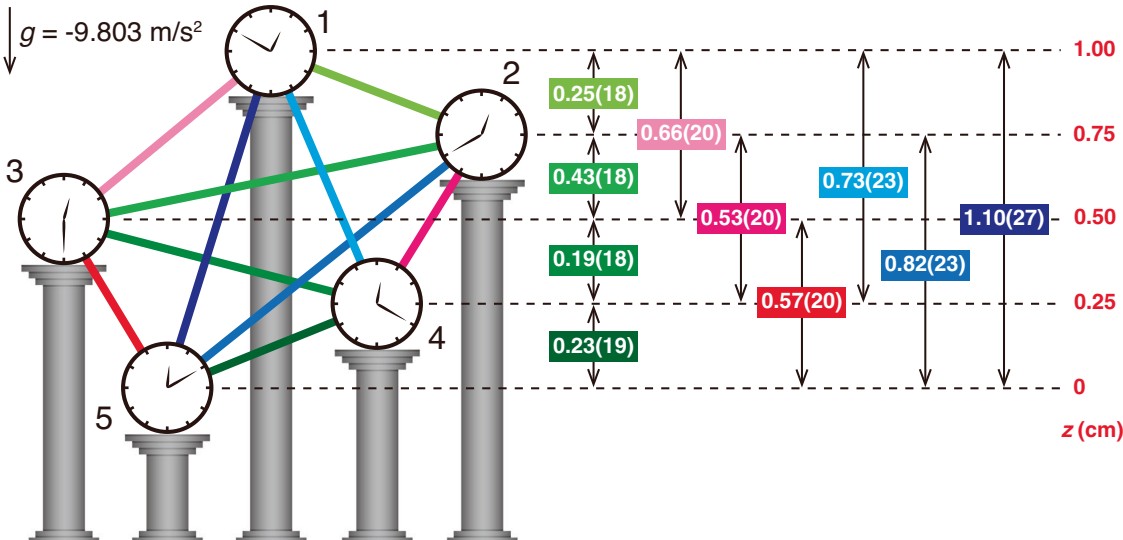

**Fig. 4 | Extracting relative height differences using relativistic gravitational potential measurements.** The relative clock height differences across the array are determined using the measured gravitational redshifts and the independently measured local gravitational acceleration $g$. The double arrows represent the extracted height difference and the associated uncertainty for each clock pair. The true clock heights are shown on the right (red values), with the lowest clock (clock 5) defined as being at a height of 0 cm. All units are in cm unless otherwise specified. The uncertainties correspond to $1\sigma$ standard deviation.

within $1\sigma$ total uncertainty. Our result is inconsistent with zero gravitational redshift at a $4.9\sigma$ confidence level and constrains deviations from the redshift predicted by general relativity to $0.13 \pm 0.23$ for mm to cm scale height differences. We demonstrate a gravitational redshift measurement resolution of 1.3 mm. Our results highlight the use of the spatially multiplexed ensemble techniques for achieving long coherence times and low differential instabilities without the need for a state-of-the-art clock laser, and demonstrate its utility for characterization of spatially varying systematic shifts in optical lattice clocks on the sub-centimeter scale and at the $10^{-19}$ level. These results represent an important milestone along the way to gravitational potential measurements at the sub-centimeter scale with optical atomic clocks[15,31–34], and explorations of the interplay between quantum mechanics and gravity[10–12].

## Methods

### Sample preparation and experimental procedure

The experimental sequence starts with laser cooling the atoms down to 1 μK temperature with standard two-stage magneto-optical trapping (MOT). Using the multiplexed ensemble loading technique with a movable one-dimensional (1D) optical lattice described and demonstrated in ref. 43, five ensembles of ultra-cold, spin-mixed $^{87}$Sr atoms are loaded into a vertical optical lattice with a depth of $130E_{rec}$, where $E_{rec}/h \approx 3.5$ kHz is the recoil energy of a lattice photon and $h$ is the Planck constant, with an equal spacing between ensembles of 0.25 cm over a total extent of 1 cm vertically. This is followed by hyperfine spin polarization into either stretched state ($|^1S_0, m_F = \pm 9/2\rangle$) and in-lattice cooling (Supplementary Note 1). The lattice is then adiabatically ramped down to the operational trap depth ($u_{op} = 15E_{rec}$), at which a series of $\pi$ pulses addressing the $^1S_0 \leftrightarrow {}^3P_0$ ($g \leftrightarrow e$) transitions prepare the atoms into $|e, m_F = \pm 3/2\rangle$. Ramsey spectroscopy is performed by interrogating the $|g, m_F = \pm 5/2\rangle \leftrightarrow |e, m_F = \pm 3/2\rangle$ clock transitions. Following the second Ramsey $\pi/2$ pulse, the lattice is adiabatically ramped back up to $130\ E_{rec}$ for read-out. The populations in the ground and excited clock states of all five ensembles are read-out in parallel with imaging pulses along the lattice axis, with scattered photons collected on a camera (Andor, iXon-888). The excitation fraction is extracted through $P = (N_e - N_{bg})/(N_g + N_e - 2N_{bg})$, where $N_g$, $N_e$ and $N_{bg}$ are the ground state population, excited clock state population, and background counts without atoms, respectively.

As described in ref. 43, for multiple ensemble preparation, we chirp the frequency of the retro-reflected lattice beam during the single-frequency stage of the narrow-line $^1S_0 \leftrightarrow {}^3P_1$ second-stage MOT. We perform four lattice movements of 0.25 cm each in order to load five spatially separated atom ensembles, and a final lattice movement of 0.5 cm in the opposite direction, which positions the ensemble array symmetrically around the lattice beam waist. The entire duration of the moving lattice portion of the loading sequence is typically less than 100 ms, with 80% of total atom number transferred efficiently from the narrow-line MOT to the five ensembles. In-lattice axial (sideband) and radial (Doppler) cooling are subsequently applied to lower the atom temperatures after lattice acceleration.

For clock interrogation, we probe the $|g, m_F = \pm 5/2\rangle \leftrightarrow |e, m_F = \pm 3/2\rangle$ transitions with a shared clock laser along the lattice axis. Synchronous Ramsey spectroscopy is performed to reject common-mode local oscillator noise. The typical Ramsey interrogation time ($T_R$) is roughly 10 s, with a dead time ($T_d$) of 2 s between interrogations for sample preparation and read-out, yielding a measurement duty cycle of 83%. Simultaneously probing five ensembles results in 10 pairwise clock comparisons for a single nuclear spin state. Combined with atom numbers ($N$) of ~2000 per ensemble and contrast ($C$) of above 80%, the typical differential instability for each pairwise comparison is below $1 \times 10^{-17}/\sqrt{\tau}$, where $\tau$ is the averaging time, consistent with the quantum projection noise (QPN) limit

$$\sigma_{QPN}(\tau) = \frac{\sqrt{2}}{2\pi f C T_R}\sqrt{\frac{T_R + T_d}{N\tau}}, \qquad (3)$$

where $f$ is the clock frequency, $\tau$ is the averaging time, and the factor of $\sqrt{2}$ assumes equal contribution from each clock.

### Ellipse phase extraction

We perform synchronous Ramsey spectroscopy with up to five ensembles (indexing 1, 2, …, 5 from top to bottom). This results in 10 pairs of clock comparisons performed simultaneously when probing transition from either nuclear spin states $|g, m_F = \pm 5/2\rangle$. For each pair of clock comparison, we plot the excitation fractions $P_i (P_j)$ of ensemble

$i(j)$ on the x(y)-axis (note that we choose the convention $j > i$). The excitation fractions are given by

$$P_i = \frac{1}{2}\left[1 + C_i \cos(\theta_L)\right],$$
$$P_j = \frac{1}{2}\left[1 + C_j \cos(\theta_L + \phi_d)\right], \qquad (4)$$

where $C_{i(j)}$ is the contrast of ensemble $i(j)$, $\theta_L$ is the common-mode laser phase, and $\phi_d$ is the differential phase which yields the differential frequency ($\delta f_{ij} = f_j - f_i$) between ensemble pair $(i, j)$ through $\phi_d = 2\pi \delta f_{ij} T_R$ for a given known Ramsey free evolution time $T_R$.

Since we are operating at Ramsey dark times well beyond the laser coherence times, $\theta_L$ is random and uniformly distributed from 0 to $2\pi$. The data randomly samples from points lying on an ellipse (with slight deviations from the ellipse due to QPN). We then fit to this ellipse using a least-squares approach[56]. To extract the differential phase[57], we rewrite the data $\{P_i, P_j\}$ (denoted as $\{x, y\}$ below) in the form of a generalized conic section

$$a_1 x^2 + a_2 xy + a_3 y^2 + a_4 x + a_5 y + a_6 = 0, \qquad (5)$$

which describes an ellipse when $a_2^2 - 4a_1 a_3 < 0$. We rewrite Eq. (4) as

$$x' = \frac{2x - 1}{C_x} = \cos(\theta_L),$$
$$y' = \frac{2y - 1}{C_y} = \cos(\theta_L + \phi_d). \qquad (6)$$

Through canceling out $\theta_L$, we have

$$\frac{4}{C_x^2} x^2 - \frac{8\cos\phi_d}{C_x C_y} xy + \frac{4}{C_y^2} y^2 + \left(\frac{4}{C_x C_y} - \frac{4}{C_x^2}\right)x + \left(\frac{4}{C_x C_y} - \frac{4}{C_y^2}\right)y$$
$$+ \left(\frac{1}{C_x^2} - \frac{2}{C_x C_y} + \frac{1}{C_y^2} - \sin^2\phi_d\right) = 0, \qquad (7)$$

which can be matched up with the coefficients $a_i$ from Eq. (5). The differential phase $\phi_d$ is then extracted using:

$$\phi_d = \cos^{-1}\left(\frac{-a_2}{2\sqrt{a_1 a_3}}\right). \qquad (8)$$

The associated Allan deviation is extracted via jackknifing technique[58], and is then fitted to a white frequency noise model with $1/\sqrt{\tau}$ scaling. Extrapolating the fit to the full averaging time yields the statistical uncertainty of the differential frequency.

In our measurements, we probe with an interleaved sequence between clock transitions with either nuclear spin state, $|g, m_F = \pm 5/2\rangle \leftrightarrow |e, m_F = \pm 3/2\rangle$. This results in 10 ellipses for transition with a single nuclear spin state, and thus 20 ellipses per measurement. A representative plot of the $|g, m_F = +5/2\rangle \leftrightarrow |e, m_F = +3/2\rangle$ transition is shown in Fig. 1b. The differential phase for each ellipse is dominated by the differential first-order Zeeman shift (Supplementary Note 3A), which is on the order of $\pm 8 \times 10^{-17}$/cm and is rejected by averaging transitions with opposite spin states.

## Data blinding protocol
To eliminate possible bias of our data taking and systematic analysis towards an expected outcome, we employ a data blinding protocol. Our data software adds a large constant offset gradient to our measurements, including the data taken for systematic evaluations and data runs taken under normal operating conditions. The blinded offset gradient is pseudo-randomly drawn from a uniform distribution spanning over $\pm 5 \times 10^{-18}$/cm, 10 times the size of the expected redshift

gradient. The blinded offset gradient is scaled by the height difference between each ensemble pair, and is then automatically added to the results from our data analysis code for ellipse phase extraction.

The blinded offset gradient was only unblinded after finalizing the corrections for all systematic effects, determining the measured value with blinded offset gradient taken under normal operating conditions, and finalizing the associated statistical and systematic uncertainties. No additional data was taken and no changes were made to the analysis, the error budget, the measured value, or the uncertainties after unblinding.

## Normal operation, data taking, unblinding and analysis
We performed 14 blinded measurement runs of gravitational redshift data under normal operating conditions over a 3-week campaign. Each run ranged in duration from 1 to 4 h, and was performed in conjunction with verification of several experimental parameters to ensure that the associated systematic effects are under control, such as the magnetic field gradient, density shift coefficients, $\delta u$, and clock and lattice beam alignments (See Supplementary Note 3). In each measurement run, the differential frequency of each ensemble pair was extracted through ellipse fitting, with the associated Allan deviation extrapolated to the full averaging time taken as the statistical uncertainty. The corrections for density shifts and bias error from the ellipse fitting are applied to each ensemble pair individually. The total uncertainty of each clock comparison is given by the quadrature sum of its statistical uncertainty and the uncertainties of systematic corrections. We analyze the measured frequency gradient using two approaches.

In the first approach, the extracted frequency differences for 10 ensemble pairs from each measurement run are plotted as a function of the height differences. A linear fit is applied to each measurement run. Many of the clock comparison pairs share a clock, e.g., pair (1, 2) and pair (2, 3) share clock 2. This means that the QPN is partially correlated between pairs, and not accounting for this would result in an underestimation of the error bar associated with the fit. To account for this, the covariance matrix is included in the fitting algorithm, where the covariance between clock pairs (a, b) and (b, c) is given by the jackknifing re-sampling approach[58]

$$\text{Cov}(\phi) = \frac{N-1}{N} \sum_i \left(\bar{\phi}_{ab}^{JK} - \bar{\phi}_{ab,\neq i}^{JK}\right)\left(\bar{\phi}_{bc}^{JK} - \bar{\phi}_{bc,\neq i}^{JK}\right), \qquad (9)$$

where $\bar{\phi}_{\neq i}^{JK}$ is the extracted phase except the $i^{\text{th}}$ data point, $\bar{\phi}^{JK}$ is the mean of $\bar{\phi}_{\neq i}^{JK}$, and $N$ is the total number of measurements.

The associated fitted slopes from 14 measurement runs, after accounting for systematic gradient corrections, are then weighted averaged, yielding a statistical uncertainty of $0.7 \times 10^{-19}$/cm, inflated by the square root of the reduced $\chi^2$ statistic, $\chi^2_{\text{red}} = 1.16$ (Fig. 3a). Upon completion of the measurements, the pseudo-randomly generated offset blinding gradient was revealed and subtracted from the measurements. The offset gradient proved to be $+3.7 \times 10^{-18}$/cm, and the measurements before and after unblinding are shown in Supplementary Fig. 8. We find a weighted mean frequency gradient of $[-12.4 \pm 0.7_{(\text{stat})} \pm 2.5_{(\text{sys})}] \times 10^{-19}$/cm, consistent with the expected redshift gradient of $-10.9 \times 10^{-19}$/cm within 1$\sigma$ total uncertainty.

In the second approach, we re-analyze the data and apply systematic corrections to each pairwise clock comparison individually over the same raw data set. For each pair, the total uncertainty is calculated as the quadrature sum of the standard error of its weighted mean, systematic uncertainties that don't scale with height difference (density shift and ellipse fitting corrections), and other systematic uncertainties that scale with height difference. The weighted averaged frequency differences of each ensemble pair are given in Supplementary Table 1. Through a final linear fit to the differential frequencies as a function of the height differences, we find a frequency gradient of

$(-11.9 \pm 2.6) \times 10^{-19}$/cm (Fig. 3b), again fully consistent with the expected redshift gradient within $1\sigma$ total uncertainty.

## Relativistic clock height difference measurements

Assuming that theory of general relativity is correct and that the gravitational redshift is given by Eq. (1), and treating the ensemble array as a network of spatially distributed clocks with unknown heights, we demonstrate relativistic gravitational potential measurement[15,32,34] using synchronous clock comparisons (Supplementary Table 1). The height difference $\Delta h$ for each clock pair is then given by

$$\Delta h = \frac{\delta f}{f} \frac{c^2}{g},\tag{10}$$

where $\delta f$ is the measured gravitational redshift, $g$ is the independently measured local gravitational acceleration ($g = -9.803$ m/s$^2$), $f$ is the clock transition frequency, and $c$ is the speed of light. The uncertainties of the extracted height differences are dominated by the systematic uncertainties of the measured gravitational redshifts.

## Data availability

The process data used in this study have been deposited on Zenodo digital repository under https://doi.org/10.5281/zenodo.8184043.

## Code availability

The code used for experimental control, data analysis, and simulation in this work are available from the corresponding author upon request.

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

## Acknowledgements

We acknowledge G.E.W. Marti, A. Jayich and T. Bothwell for fruitful discussions and insightful comments on the manuscript. We acknowledge technical contributions from B.N. Merriman, H. Li, V. Lochab and N. Ranabhat. We are particularly grateful to B. Tikoff, E. M. Nelson and C. Ruggles from the Department of Geoscience at the University of Wisconsin-Madison for performing measurements of the gravitational acceleration in our laboratory. This work was supported by the NIST Precision Measurement Grants program, the Northwestern University Center for Fundamental Physics and the John Templeton Foundation through a Fundamental Physics grant, the Wisconsin Alumni Research Foundation, a Packard Fellowship for Science and Engineering, a Sloan Research Fellowship, the Army Research Office through agreement number W911NF-21-1-0012, and the National Science Foundation under Grant No. 2143870.

## Author contributions

X.Z., J.D., M.C.C, H.-M.L. and S.K. all contributed to carrying out the experiments, the data analysis, and to the writing of the manuscript.

## Competing interests

The authors declare no competing interests.
