## [Peer Review File · Nature Communications]

A lab-based test of the gravitational redshift with a miniature clock networkEditorial Note: This manuscript has been previously reviewed at another journal that is not operating a transparent peer review scheme. This document only contains reviewer comments and rebuttal letters for versions considered at *Nature Communications*. Mentions of the other journal have been redacted.

Reviewers' comments:

Reviewer #2 (Remarks to the Author):

The present manuscript was transferred from [redacted] to Nature Communications and includes a number of changes to address points from the previous reviews.

Altogether, my previous comments related to geodetic topics have been addressed very well in the point-by-point responses as well as in the new manuscript itself.

I have only a very small comment related to a sentence on page 3: “With optical clocks now reaching instabilities and inaccuracies at the level of 10⁻¹⁸ and below [8,37–40], they are becoming a sensitive probe of the point to point geopotential at the sub-centimeter scale, where they are expected to complement other methods of geodesy that do not make use of spatial averaging [8,12–16].” – I recommend to remove the last part “that do not make use of spatial averaging”, because it is unclear what it means and it is also unnecessary.

Altogether, I think that the new manuscript is greatly improved over the initial version, and I find no more critical statements with respect to geodetic issues. The paper is very well written and the redshift measurements at the mm level in height are very impressive. However, as stated in my previous review, I cannot really overlook the importance of the physics parts of the study, but, assuming that there is enough innovative material, I recommend to publish the paper in Nature Communications.

Reviewer #3 (Remarks to the Author):

Dear Editor(s),

Dear Authors,

I continue to serve on a reviewer following the previous round of review at another Nature journal. First of all, I would like to thank the Authors for sincere response to the comments I raised at the previous round.

I am writing again that this manuscript written by X. Zeng et al. is very interesting and clear, and this work would be expected to provide an important contribution to the relativistic geodesy, which is one of the expected applications of accurate optical clocks, therefore I think that the readers at least in geodesy community as well as metrology would be interested in this work, and the method presented here would be expected to be a new platform for the study of new physics, gravitational wave detection, and the union of general relativity and quantum mechanics. The Authors also very carefully evaluate potential systematics for their differential clock comparisons, that is one of the updated points from the Authors' previous work X. Zheng, et al., "Differential clock comparisons with a multiplexed optical lattice clock", Nature 602, 425-430 (2022) (reference [49] in the manuscript).

The manuscript itself is very carefully and scientifically well written, but there is the point which I would like the Authors to reconsider and improve as I will note in my comment below. I would like to support its publication in Nature Communications assuming the Authors can fully address my following concern.

To my understanding, the important highlights of this work are

- 1) a demonstration of a laboratory-based test of the gravitational redshift using the millimetre to centimetre-level clock network with synchronous comparisons,
- 2) achievement of the accuracy of the 10^{-19} level for its test with a blind scheme, systematic evaluation of all the sources of differential clock shifts at the 10^{-19} level.

As written in the abstract of the submitted manuscript, the test of the gravitational redshift has been performed by comparing separated atomic clocks over a wide range of height differences (reference [2-7] in the manuscript). The lab-based test from mm to cm level in

the manuscript could give new insights to previous knowledge. However, this achievement should be compared to that of the prior work T. Bothwell et al., “Resolving the gravitational redshift across a millimetre-scale atomic sample”, *Nature* 602, 420-424 (2022) (reference [7] in the manuscript). The prior work has already demonstrated a precise test of the gravitational redshift across a single millimetre scale atomic ensemble in a vertical 1D optical lattice. In this work, furthermore, T. Bothwell et al. employed the clock laser, which has a very narrow linewidth of the 8 mHz, which is two orders of magnitude narrower than that in the Authors’ work, that should be great advantage to perform a precise measurement. Nevertheless, as far as I can see, T. Bothwell et al. apply the synchronous comparison between two uncorrelated regions of their atomic cloud, that could suppress the common noise of the clock laser. Therefore, I think this concept, the suppression of the local oscillator noise with synchronous measurement in the same optical lattice for the test of gravitational redshift, is very similar to that in the Authors’ work even though there is a difference between employing Rabi and Ramsey spectroscopy. Incidentally, the concept of the synchronous comparison itself has been demonstrated by M. Takamoto et al., “Frequency comparison of optical lattice clocks beyond the Dick limit”, *Nat. Photonics* 5, 288-292 (2011). From this perspective, I would say there would not be so much impact in the Authors’ work comparing to that by T. Bothwell et al in terms of the point-1). And, T. Bothwell et al. reports their result of the gradient of $-1.28(27) \times 10^{-19} \text{ mm}^{-1}$, that seems compare favorably with Authors’ result. Therefore, it is not very clear for me that the current manuscript would report a significant achievement in terms of the point-2). However, the Authors very carefully evaluate the spatially dependent systematics to extend the test of the gravitational redshift to the larger height range, that the difference from the prior work by T. Bothwell et al. Furthermore, the Authors adopt a blinded offset during data taking and systematic evaluation, while Bothwell et al. did not. This scheme is indeed a different point from the prior work, and its concept could increase reliability in their systematic evaluation and data taking since the bias by their prior expectation could be removed. However, I am still not quite sure if this adoption would have an impact worthy of the publication in *Nature Communications*.

Additionally, regarding the point-1), the current manuscript should be compared to the Authors’ prior work X. Zheng, et al., “Differential clock comparisons with a multiplexed optical lattice clock”, *Nature* 602, 425-430 (2022) (reference [49] in the manuscript). The

scheme using the multiplexed optical lattice clock could be expected to provide a promising platform to search for new physics and other applications such as the relativistic geodesy presented in this manuscript. In the submitted manuscript, the Authors perform a comprehensive, blinded, systematic evaluation of the all sources of differential clock shift at the 10^{-19} level to leverage the test of the gravitational redshift at a new scale. However, this powerful scheme itself has been already demonstrated in the Authors' previous work in terms of building the initial apparatus, or demonstrating its basic operating principles, therefore I think the originality of the submitted work would not be very rich enough to be published in Nature Communications.

I am sure that the topic itself is very impressive for metrology and geodesy community, and there is scientific rigor in the manuscript, but from the points of view of conceptual and technological advance, I am still not entirely sure whether the Editors regard this work as being new and having an impact enough for the publication in Nature Communications. I would say the Authors' answer to the comments I raised before and its relevant revision in the manuscript would not be still satisfactorily addressed on this point, and I would like to recommend submitting the manuscript to another journal if the Authors could not fully address this point. However, if the Editors decided that this manuscript could deserve publication, I would like the Authors to address a minor comment as below.

I would like to know the more details on the evaluation of the blackbody radiation (BBR) gradient shift. The Authors heat up either the top or bottom stainless steel flange of the science chamber to make temperature difference between the top and bottom viewports and to evaluate BBR gradient along the lattice axis since the viewports are mainly responsible for the BBR gradient. Then, they measure the frequency differences of the 10 ensemble pairs simultaneously to obtain the results shown in Extended Data Fig. 3. After that, they did liner fits to the results of BBR shift differences between the pairs as a function of temperature difference between the viewports even though the BBR shift theoretically depends on the fourth power of the temperature. I am wondering if the Authors would be considering that the linear fit here make sense because the distances between the atomic ensemble pairs are at longest 1 cm and the temperature differences between them are the largest 2 K. Is my understanding correct?

We thank reviewers #2 and #3 for their careful reading of our revised manuscript, and for their insightful and constructive comments. Below we provide point-by-point responses to each of the reviewers' comments (original comments in blue italics):

Referee #2:

1) The present manuscript was transferred from [redacted] to Nature Communications and includes a number of changes to address points from the previous reviews.

Altogether, my previous comments related to geodetic topics have been addressed very well in the point-by-point responses as well as in the new manuscript itself.

We thank the referee for their positive feedback and are gratified that they find our revisions to the manuscript appropriately addressed the comments related to geodetic topics.

2) I have only a very small comment related to a sentence on page 3: "With optical clocks now reaching instabilities and inaccuracies at the level of 10⁻¹⁸ and below [8,37–40], they are becoming a sensitive probe of the point to point geopotential at the sub-centimeter scale, where they are expected to complement other methods of geodesy that do not make use of spatial averaging [8,12–16]." – I recommend to remove the last part "that do not make use of spatial averaging", because it is unclear what it means and it is also unnecessary.

We thank the referee for this advice and have followed it. The last part of that sentence has now been removed.

3) Altogether, I think that the new manuscript is greatly improved over the initial version, and I find no more critical statements with respect to geodetic issues. The paper is very well written and the redshift measurements at the mm level in height are very impressive. However, as stated in my previous review, I cannot really overlook the importance of the physics parts of the study, but, assuming that there is enough innovative material, I recommend to publish the paper in Nature Communications.

We are glad that the referee finds our manuscript impressive and well written, and believe that there is more than sufficient innovative material to justify publication in *Nature Communications*, as validated by Referee #1's strong recommendation for publication in [redacted], as well as the by the positive remarks of all three referees regarding the impressive nature of the results and clarity and quality of the manuscript.

Referee #3:

1) I continue to serve on a reviewer following the previous round of review at another Nature journal. First of all, I would like to thank the Authors for sincere response to the comments I raised at the previous round.

I am writing again that this manuscript written by X. Zeng et al. is very interesting and clear, and this work would be expected to provide an important contribution to the relativistic geodesy, which is one of the expected applications of accurate optical clocks, therefore I think that the readers at least in geodesy community as well as metrology would be interested in this work, and the method presented here would be expected to be a new platform for the study of new physics, gravitational wave detection, and the union of general relativity and quantum mechanics. The Authors also very carefully evaluate potential systematics for their differential clock comparisons, that is one of the updated points from the Authors' previous work X. Zheng, et al., "Differential clock comparisons with a multiplexed optical lattice clock", Nature 602, 425-430 (2022) (reference [49] in the manuscript).

We are gratified that the referee finds our manuscript interesting, clear, and important, and thank them for the kind words.

2) The manuscript itself is very carefully and scientifically well written, but there is the point which I would like the Authors to reconsider and improve as I will note in my comment below. I would like to support its publication in Nature Communications assuming the Authors can fully address my following concern.

To my understanding, the important highlights of this work are

- (1) a demonstration of a laboratory-based test of the gravitational redshift using the millimetre to centimetre-level clock network with synchronous comparisons,*
- (2) achievement of the accuracy of the 10^{-19} level for its test with a blind scheme, systematic evaluation of all the sources of differential clock shifts at the 10^{-19} level.*

As written in the abstract of the submitted manuscript, the test of the gravitational redshift has been performed by comparing separated atomic clocks over a wide range of height differences (reference [2-7] in the manuscript). The lab-based test from mm to cm level in the manuscript could give new insights to previous knowledge. However, this achievement should be compared to that of the prior work T. Bothwell et al., "Resolving the gravitational redshift across a millimetre-scale atomic sample", Nature 602, 420-424 (2022) (reference [7] in the manuscript). The prior work has already demonstrated a precise test of the gravitational redshift across a single millimetre scale atomic ensemble in a vertical 1D optical lattice. In this work, furthermore, T. Bothwell et al. employed the clock laser, which has a very narrow linewidth of the 8 mHz, which is two orders of magnitude narrower than that in the Authors' work, that should be great advantage to perform a precise measurement. Nevertheless, as far as I can see,

T. Bothwell et al. apply the synchronous comparison between two uncorrelated regions of their atomic cloud, that could suppress the common noise of the clock laser. Therefore, I think this concept, the suppression of the local oscillator noise with synchronous measurement in the same optical lattice for the test of gravitational redshift, is very similar to that in the Authors' work even though there is a difference between employing Rabi and Ramsey spectroscopy. Incidentally, the concept of the synchronous comparison itself has been demonstrated by M. Takamoto et al., "Frequency comparison of optical lattice clocks beyond the Dick limit", Nat. Photonics 5, 288-292 (2011). From this perspective, I would say there would not be so much impact in the Authors' work comparing to that by T. Bothwell et al in terms of the point-1). And, T. Bothwell et al. reports their result of the gradient of $-1.28(27) \times 10^{-19} \text{ mm}^{-1}$, that seems compare favorably with Authors' result. Therefore, it is not very clear for me that the current manuscript would report a significant achievement in terms of the point-2).

However, the Authors very carefully evaluate the spatially dependent systematics to extend the test of the gravitational redshift to the larger height range, that the difference from the prior work by T. Bothwell et al. Furthermore, the Authors adopt a blinded offset during data taking and systematic evaluation, while Bothwell et al. did not. This scheme is indeed a different point from the prior work, and its concept could increase reliability in their systematic evaluation and data taking since the bias by their prior expectation could be removed. However, I am still not quite sure if this adoption would have an impact worthy of the publication in Nature Communications.

We have carefully considered the point made by the referee, and we respectfully, but strongly, disagree with the referee's assessment, and we would like to make several points in response. First, we wish to note that much of this work was performed contemporaneously to Bothwell et al. and was not inspired by it in any way. We completed this work and initially posted it to the arXiv roughly 6 months after Bothwell et al was published. In addition, we would also like to point out that Bothwell et al. was published in *Nature* and featured on the cover, illustrating the level of interest in similar results. We believe that being the first to publish an important result is not the only figure of merit that determines a works ultimate impact or significance, and that there is value both in the careful replication of results and especially in the demonstration of complementary and novel approaches. We believe our work is both highly complementary to Bothwell et al. and also significantly different from it and the other works mentioned by the referee (as we emphasize below), and we note that Referee #1 apparently agreed with this assessment and strongly recommended publication of our manuscript in [redacted]. We would kindly ask the referee to reconsider their assessment in light of this perspective and our responses below.

- a. *"Therefore, I think this concept, the suppression of the local oscillator noise with synchronous measurement in the same optical lattice for the test of gravitational redshift, is very similar to that in the Authors' work"*

We respectfully disagree with this assessment. Bothwell et al. did not probe beyond their clock laser coherence time (which is on the order of 10 s, compared to the Rabi interrogation time of 3 s). While the interrogation time (typically 8 s) in our work is almost two orders of magnitude longer than our laser coherence times (100 ms).

- b. *“... even though there is a difference between employing Rabi and Ramsey spectroscopy”*

We believe this is an extremely important point that is being downplayed by the referee. While most existing optical lattice clocks employ Rabi spectroscopy, to the best of our knowledge, our work represents the first demonstration of full systematic evaluations at 10^{-19} level with synchronous Ramsey spectroscopy that probes beyond the laser coherence limit, and thus represents a significant advance over previous works. It is not possible to perform Rabi spectroscopy with interrogation times beyond the coherence time of the local oscillator, meaning that the experiments performed by Bothwell et al could likely only currently be performed by the two research groups in the world that have access to clock lasers with <10 mHz instantaneous linewidths, while the synchronous Ramsey spectroscopy technique we demonstrate here can be employed by dozens of research groups around the world. We note that although synchronous Ramsey spectroscopy was performed in Takamoto et al. Nat. Photon. (2020) to measure the redshift at 450 m, they did not probe beyond the laser coherence limit in that work. Our results demonstrate for the first time that the achievable accuracy and precision of tests of fundamental physics with optical lattice clocks do not always have to be limited by the quality of the clock laser used for the experiments.

- c. *“Incidentally, the concept of the synchronous comparison itself has been demonstrated by M. Takamoto et al., “Frequency comparison of optical lattice clocks beyond the Dick limit”, Nat. Photonics 5, 288-292 (2011).”*

The reviewer is correct that the concept of synchronous optical clock comparisons was demonstrated in Takamoto et al. (2011). However, as in Bothwell et al, that work did not demonstrate interrogation times beyond the coherence time of the clock laser, and also did not use those measurements to perform a systematic evaluation at the level of 10^{-19} . We have added a reference to Takamoto et al. to the revised manuscript along with added context regarding what differentiates our work from past works that demonstrated synchronous comparisons.

- d. *“From this perspective, I would say there would not be so much impact in the Authors’ work comparing to that by T. Bothwell et al in terms of the point-1). Therefore, it is not very clear for me that the current manuscript would report a significant achievement in terms of the point-2).”*

Again, we wish to emphasize that to our knowledge, our work is the first to demonstrate the use of synchronous differential comparisons well beyond the coherence time of the local oscillator to perform a precision measurement of a physical quantity of interest. We have revised the manuscript to provide this context more clearly.

- e. *“And, T. Bothwell et al. reports their result of the gradient of $-1.28(27) \times 10^{-19} \text{ mm}^{-1}$, that seems compare favorably with Authors’ result. Therefore, it is not very clear for me that the current manuscript would report a significant achievement in terms of the point-2). However, the Authors very carefully evaluate the spatially dependent systematics to extend the test of the gravitational redshift to the larger height range, that the difference from the prior work by T. Bothwell et al. Furthermore, the Authors adopt a blinded offset during data taking and systematic evaluation, while Bothwell et al. did not. This scheme is indeed a different point from the prior work, and its concept could increase reliability in their systematic evaluation and data taking since the bias by their prior expectation could be removed. However, I am still not quite sure if this adoption would have an impact worthy of the publication in Nature Communications.”*

While it is true that Bothwell et al. reported a similar measurement level (20% uncertainty of a 1 mm gravitational redshift) as in our work (20% uncertainty of a 1 cm redshift), there are several key differences between the two works:

- i) Our rack-mounted local oscillator is nearly two orders of magnitude worse than the state-of-the-art, cryogenic local oscillator employed in Bothwell et al., while achieving a comparable level of measurement uncertainty. This is encouraging for clock applications involving transportable local oscillators or for the wider adoption of these techniques by groups around the world that do not have access to the local oscillator used by Bothwell et al.
- ii) While the redshift in our work (10^{-18} @ 1 cm) is an order of magnitude larger than the redshift in Bothwell (10^{-19} @ 1 mm), we note that, because of the increased spatial scale, the systematic effects require additional efforts to control and characterize. We observed several systematic shifts, such as the differential tensor Stark shift and differential BBR shift, that weren’t observed in the work of Bothwell et al.
- iii) As the reviewer pointed out, our work employs a blinded frequency gradient offset to remove any unconscious bias towards an expected outcome, which was not used in the work of Bothwell et al. We believe the use of a blind adds an additional degree of confidence that we have not overlooked any potentially offsetting systematic effects, and helps to validate the novel techniques we used to characterize differential systematics at the 10^{-19} level.

5) Additionally, regarding the point-1), the current manuscript should be compared to the Authors’ prior work X. Zheng, et al., “Differential clock comparisons with a multiplexed optical lattice clock”, Nature 602, 425-430 (2022) (reference [49] in the

manuscript). The scheme using the multiplexed optical lattice clock could be expected to provide a promising platform to search for new physics and other applications such as the relativistic geodesy presented in this manuscript. In the submitted manuscript, the Authors perform a comprehensive, blinded, systematic evaluation of the all sources of differential clock shift at the 10^{-19} level to leverage the test of the gravitational redshift at a new scale. However, this powerful scheme itself has been already demonstrated in the Authors' previous work in terms of building the initial apparatus, or demonstrating its basic operating principles, therefore I think the originality of the submitted work would not be very rich enough to be published in Nature Communications.

We respectfully disagree with the referee's assessment. As the Referee states, the major difference between this work and our prior work (Ref. [49]) is that we tested the gravitational redshift at mm-scale with the platform by performing a full systematic characterization at the 10^{-19} level, which had not been demonstrated in Ref. [49]. The careful and thorough evaluation of systematic effects is by far the hardest and most time-consuming part of any precision test of fundamental physics, and the proper accounting of all possible systematic effects at the 10^{-19} level with an applied blind to prevent bias is no small feat. Even following our prior work, it was not at all clear that correlated Ramsey spectroscopy could be used to fully evaluate all possible differential systematic effects, and in the course of this work we developed a number of new techniques and analysis tools to perform the systematic evaluation, all of which are detailed in the manuscript and the supplement. As a result, we also observed several new systematic effects that have not previously been observed, including in Bothwell et al., and which will likely become relevant to absolute clocks as their precision and accuracy continues to improve.

6) I would like to know the more details on the evaluation of the blackbody radiation (BBR) gradient shift. The Authors heat up either the top or bottom stainless steel flange of the science chamber to make temperature difference between the top and bottom viewports and to evaluate BBR gradient along the lattice axis since the viewports are mainly responsible for the BBR gradient. Then, they measure the frequency differences of the 10 ensemble pairs simultaneously to obtain the results shown in Extended Data Fig. 3. After that, they did liner fits to the results of BBR shift differences between the pairs as a function of temperature difference between the viewports even though the BBR shift theoretically depends on the fourth power of the temperature. I am wondering if the Authors would be considering that the linear fit here make sense because the distances between the atomic ensemble pairs are at longest 1 cm and the temperature differences between them are the largest 2 K. Is my understanding correct?

We thank the referee for bringing up this important point, which is an excellent example of a new differential systematic effect that had not previously been studied or observed, and was not observed in Bothwell et al. The referee is correct, we find that to first order the BBR gradient results in linear scalings of the frequency difference between two

atomic ensembles with both the temperature difference and the spatial separation of the ensembles. In response to the referee's question we have added a detailed derivation to the supplement in the revised manuscript. As the referee suggests, this is appropriate here because the absolute temperature (~ 300 K) is a factor of >150 x larger than the temperature difference (< 2 K), and the pairwise separations of the atom ensembles (0.25, 0.50, 0.75 and 1 cm) are more than an order of magnitude smaller than the distance between the top and bottom viewports (> 15 cm). We also find that this nicely agrees with the results of our experiments. The typical reduced χ^2 in the linear fit is about 1.7, and the uncertainty of the fitted slope (BBR sensitivity for each ensemble pair) has been inflated by the square root of the reduced χ^2 , and is accounted for in the BBR shift evaluation. We have updated the revised manuscript with this added context and these details.

Summary of changes made to the manuscript:

1. The sentence "*that do not make use of spatial averaging*" on Page 3 is removed.
2. A discussion, "*To the best of our knowledge...*", is added to the main text in Page 5 to emphasize the difference between the synchronous Ramsey spectroscopy employed in this work and the Rabi spectroscopy used in the works of Bothwell et al. and Takamoto et al.
3. A theoretical derivation of the linear scaling of the differential BBR shift is included in the Supplement (Page 33).

We trust that these changes thoroughly address the above comments and concerns, and thank all of the reviewers once again for their time, effort, and thoughtful consideration of this manuscript.

REVIEWERS' COMMENTS

Reviewer #3 (Remarks to the Author):

Dear Editor(s),

Dear Authors,

I thank the Author for sincerely responding to my concerns, and I would like to deeply apologize for my misunderstanding. I now have a clear understanding of the synchronous Ramsey spectroscopy thanks to the Authors' patient explanations from "a" to "c" in the Authors' response. As Authors claimed, that scheme should be highly evaluated in the point that the Authors could probe beyond the laser coherence time. I realized this was a different point from previous works performed by other groups as well as T. Bothwell et al. However, we should not overlook the fact that this scheme, differential clock comparisons by the synchronous Ramsey spectroscopy that probes beyond the laser coherence time, itself was correctly evaluated to be already published in Nature Ref[50] (X. Zheng et al., Nature 602, 425-430 (2022)) as the Authors' previous work. This might be the point Referee #2 concerns in 3) of their comments. Therefore, as the Authors mention in ii) and iii) in "e" and 5), the difference points of this work from the work by Bothwell et al. and the Authors' prior work (Ref[50]) are careful evaluations of systematic effects with the synchronous Ramsey spectroscopy and the blinded offset scheme to reach the accuracy of the 10^{-19} level for the test of the gravitational redshift. I really agree with that such an evaluation would be the hardest and most time-consuming part of any precision measurements, therefore I honestly respect such the work, but I have to say that such an effort does not necessarily have an enough impact to meet the highly impacted journal like the Nature Communications. I guess such an issue is where a researcher of a precision measurement such a metrology always struggles.

Although it becomes repetitive, as the Authors mentioned in "e" and 5) in their response, by employing the synchronous Ramsey spectroscopy, which is already highly evaluated to be published in Nature (Ref[50]) and the blinded frequency gradient offset, which is not used in the previous works, in order to avoid any unconscious bias in the test of the gravitational redshift, the Authors reached the similar level as that of the work demonstrated by Bothwell et al. (Ref[7]). This achievement also required the Authors' effort to evaluate several

systematic shifts, such as the differential tensor Stark shift and differential BBR shift, that weren't observed in the work by Bothwell et al. As I mentioned in "e", this point is indeed a difference from the prior works, and its concept could increase reliability in their systematic evaluation and data taking, however I am not quite sure if this adoption would have an impact worthy of the publication in Nature Communications.

I can expect that readers in metrology and geodesy community would be very interested in the topic itself, and the manuscript itself is very carefully and scientifically well written, however I am still not entirely sure whether the Editor(s) regards this work as being enough new or impact for the publication in Nature Communications.

Regarding 6) in the Authors' response, my previous concern about BBR shift have been addressed very well in the revised manuscript.